# The Prostaglandin E2 Receptor EP4 Promotes Vascular Neointimal Hyperplasia through Translational Control of Tenascin C via the cAPM/PKA/mTORC1/rpS6 Pathway

**DOI:** 10.3390/cells11172720

**Published:** 2022-08-31

**Authors:** Hu Xu, Bingying Fang, Chengzhen Bao, Xiuhui Mao, Chunhua Zhu, Lan Ye, Qian Liu, Yaqing Li, Chunxiu Du, Hang Qi, Xiaoyan Zhang, Youfei Guan

**Affiliations:** 1Advanced Institute for Medical Sciences, Dalian Medical University, Dalian 116041, China; 2Health Science Center, East China Normal University, Shanghai 200241, China

**Keywords:** vascular stenosis, prostanoid, tenascin C, EP4, gene knockout

## Abstract

Prostaglandin E2 (PGE2) is an important metabolite of arachidonic acid which plays a crucial role in vascular physiology and pathophysiology via its four receptors (EP1-4). However, the role of vascular smooth muscle cell (VSMC) EP4 in neointimal hyperplasia is largely unknown. Here we showed that VSMC-specific deletion of EP4 (VSMC-EP4) ameliorated, while VSMC-specific overexpression of human EP4 promoted, neointimal hyperplasia in mice subjected to femoral artery wire injury or carotid artery ligation. In vitro studies revealed that pharmacological activation of EP4 promoted, whereas inhibition of EP4 suppressed, proliferation and migration of primary-cultured VSMCs. Mechanically, EP4 significantly increased the protein expression of tenascin C (TN-C), a pro-proliferative and pro-migratory extracellular matrix protein, at the translational level. Knockdown of TN-C markedly suppressed EP4 agonist-induced VSMC proliferation and migration. Further studies uncovered that EP4 upregulated TN-C protein expression via the PKA/mTORC1/Ribosomal protein S6 (rpS6) pathway. Together, our findings demonstrate that VSMC EP4 increases TN-C protein expression to promote neointimal hyperplasia via the PKA-mTORC1-rpS6 pathway. Therefore, VSMC EP4 may represent a potential therapeutic target for vascular restenosis.

## 1. Introduction

Occlusive atherosclerosis of coronary, carotid, or other peripheral arteries seriously threatens human health and imposes a huge personal and societal healthcare burden [1]. Currently, the conventional approaches to restoring blood flow of the blocked coronary, carotid, or other peripheral arteries are percutaneous vascular interventions, including balloon angioplasty and stenting [2]. However, the frequent in-stent restenosis remains a critical clinical challenge [3]. It has been well known that the pathological feature underlying vascular restenosis is neointimal hyperplasia. The proliferation and migration of vascular smooth muscle cells (VSMCs) are two major pathological processes involved in the formation of neointimal hyperplasia, which are also currently major therapeutic targets for vascular restenosis [4,5,6,7]. In response to vascular injury, the phenotypically altered VSMCs migrate from the tunica media to the intima, where they proliferate and secrete a variety of proinflammatory cytokines and chemokines and abundant extracellular matrix proteins to promote the formation of the neointima. Neointima hyperplasia together with vascular remodeling eventually leads to vascular stenosis [8].

Prostaglandin E2 (PGE2), a crucial arachidonic acid metabolite, is generated from a sequential enzymatic process including two cyclooxygenases (COX1 and COX-2) and three terminal prostaglandin E synthases (mPGES-1, mPGES-2, and cPGES). Among these enzymes, the involvement of COX-2 and mPGES-1 in neointimal hyperplasia after vascular injury has been extensively investigated [9,10]. It has been reported that COX-2-derived PGE2 promotes injury-induced vascular restenosis in mice [11] and the COX2 inhibitor celecoxib can reduce the formation of the neointima in rats [12]. Consistently, celecoxib has been shown to effectively reduce target-lesion revascularization in coronary artery disease patients with stent implantation [13]. However, the adverse cardiac events of non-fatal myocardial infarction and cardiac death were significantly higher in patients receiving celecoxib treatment, raising a concern about increased risk for thrombosis and hypertension [14,15]. Similarly, global deletion of mPGES-1 restrained VMSC proliferation and then reduced wire-induced neointimal formation via the suppression of PGE2 and augmentation of PGI2 [9]. However, phenotypic divergence was repeatedly reported when mPGES-1 was selectively knocked out among distinct cell types. Selective deletion of mPGES-1 in both endothelial cells (ECs) and VSMCs enhanced, while myeloid cell mPGES-1 deletion decreased, neointimal hyperplasia in response to wire injury [10]. Similar to the hypertensive effect of systemic inhibition of COX-2, global mPGES-1 gene deletion also resulted in enhanced vasopressor response to angiotensin II (Ang II) [16]. Therefore, although a large body of evidence supports the involvement of PGE2 in vascular neointimal formation, systemic blockade of COX-2 or mPGES-1 may lead to many severe adverse cardiac events.

PGE2 exerts its biological effects via four G protein-coupled receptors, designated EP1, EP2, EP3, EP4 [17]. Increasing evidence suggests that the PGE2-EPs system plays a vital role in vascular homeostasis and remodeling [11,18,19]. It has been previously reported that EP2 ameliorates, while EP3 facilitates, neointimal formation in a murine model of vascular restenosis [11,18]. Activation of endothelial EP4 promotes vasodilation and reendothelialization to reduce blood pressure and inhibit neointimal formation [20,21]. Nevertheless, given a critical role of VSMCs in vascular homeostasis and the pathogenesis of neointimal hyperplasia [19,22,23,24], it is anticipated that VSMC EP4 may play an important role in the pathogenesis of vascular restenosis.

Tenascin-C (TN-C) is a pro-proliferative and promigratory extracellular matrix glycoprotein that is tightly associated with tissue injury and repair [25]. Elevated expression of TN-C is found in the medial and neointimal VSMC layers after vascular injury, which provides a scaffold where proliferating VSMCs can migrate to form the neointima [26]. A previous study showed that TN-C directly contributes to neointimal formation [27]. More importantly, reduced TN-C expression was identified as a crucial down-stream regulator responsible for attenuated neointimal hyperplasia after mPGES-1 deletion [9]. However, the effect and mechanism of EP4 on TN-C expression in VSMCs remain largely unknown.

In the present study, we hypothesized that EP4 attenuates neointimal hyperplasia via regulating TN-C. By generating VSMC-specific EP4 deletion or overexpression mice, we investigated the VSMC-specific effects of EP4 on in vivo neointimal formation following artery injury. We found that the loss of EP4 specifically in VSMCs significantly attenuated ligation- and wire injury-induced neointimal hyperplasia. In contrast, selective overexpression of human EP4 gene in VSMCs led to aggravated neointimal growth. We further showed that activation of EP4 in cultured VSMCs up-regulated TN-C protein expression at the translational level via the PKA/mTORC1/rpS6 pathway. Taken together, our findings demonstrate that the VSMC EP4 is a critical player in the pathogenesis of neointimal formation and may represent a promising target for the therapeutic intervention of vascular restenosis.

## 2. Materials and Methods

### 2.1. Animals

The male mice used in this study were 12 to 16 weeks old and weighted between 22–25 g with a C57BL/6 background. The VSMC-EP4^−/−^ and VSMC-hEP4 Tg mice were generated as previously described [19,28]. All animal care and experimental procedures were reviewed and approved by the Animal Care and Use Review Committee of Dalian Medical University (AEE18046). The study conformed to the Guide for the Care and Use of Laboratory Animals published by the US National Institutes of Health.

### 2.2. Femoral Artery Wire Injury and Adenovirus-Mediated Gene Delivery

Wire-mediated vascular injury was performed in the left femoral arteries of mice by endoluminal passage of an angioplasty guide wire as previously described [29]. The sham-operated uninjured right femoral arteries served as controls. In brief, mice were anesthetized with intraperitoneal injection of 1% pentobarbital sodium (50 mg/kg). Left femoral arteries were exposed by a longitudinal groin incision and monitored under a surgical microscope. After the distal portion of the artery was encircled with a 4–0 nylon suture, a vascular clamp was placed proximally at the level of the inguinal ligament, and a guide wire 0.38 mm in diameter (Micro Therapeutics, Irvine, CA, USA) was introduced into the arterial lumen through an arteriotomy made in the distal portion of the muscular branch artery. After release of the clamp, the guide wire was advanced to the level of the arterial bifurcation and was left in place for 1 min to denude and dilate the artery. Then, the wire was removed, and the silk suture was looped at the proximal portion of the muscular branch artery. Blood flow in the femoral artery was restored by releasing the sutures placed in the proximal and distal femoral portions. For adenovirus-mediated gene delivery, after wire injury, 5 × 10^8^ pfu of adenoviruses carrying EP4 and GFP diluted in a total volume of 20 μL saline were delivered to left and right injured segments, respectively. After incubating for 20 min, the skin incision was closed with a 4–0 silk suture. Animals were monitored as routine after surgery.

### 2.3. Carotid Artery Ligation

The mice were anesthetized with an intraperitoneal injection of 1% pentobarbital sodium (50 mg/kg). Left and right common carotid arteries were exposed after a midline incision of the neck. The left common carotid artery of each mouse was ligated with a 4–0 silk suture just proximal to its bifurcations. The sham-operated uninjured right common carotid artery served as control.

### 2.4. Morphometric and Immunohistological Analysis

At the indicated time points after the injury, the mice were euthanized by CO_2_, then were perfused with saline. For the morphometry, after being immersed in 4% PFA for 24 h (4 °C), the arteries were embedded in optimal cutting temperature (OCT) compound (Sakura Fine technical, Tokyo, Japan) and stored at −80 °C. OCT-embedded arteries were cut in 5 μM serial cross sections. Within the injured distal region (2000 μM) of the left femoral artery, cross sections were analyzed in regular intervals of 100 μM to calculate a vessel mean for each parameter. Sections from the distal part of the contralateral right femoral artery (i.e., within ~2000 μM distance to the branching of the profunda femoris artery) were analyzed as an internal control. In ligated left common carotid arteries, cross sections from a predefined proximal distance from the ligation site (500 μM) were analyzed. In sham-operated right common carotid arteries, sections within 500 μM distance to the bifurcation of internal and external carotid artery were analyzed. Corresponding sections were stained with hematoxylin and eosin (H&E) and subjected to immunohistochemical analysis.

The intima and media areas were measured by computerized morphometry with Image-Pro Plus 6.1 (Media Cybernetics, Bethesda, MD, USA). Neointimal hyperplasia was defined as the formation of a neointimal layer within the internal elastic lamina (IEL). Medium area was calculated as the area encircled by the external elastic lamina (EEL) minus the area encircled by IEL. The intima-to-media ratio was calculated as the intimal area divided by the medial area. In addition, carotid arterial slices on separate glass slides were processed for the immunohistochemical analysis of EP4, PCNA, TN-C. The primary rabbit polyclonal antibody against EP4 (1:200; Cayman, Ann Arbor, MI, USA), PCNA (1:1000, Abcam, Cambridge, UK), or TN-C (1:200, Abcam, Cambridge, UK) was applied and left overnight at 4 °C. Then, the sections were sequentially treated with appropriate secondary antibodies (ZSGB-BIO, Beijing, China) for 1 h at room temperature and were visualized with DAB staining kit (ZSGB-BIO, Beijing, China).

### 2.5. RNA Extraction and Real-Time PCR

Artery segments were quickly dissected and snap-frozen in liquid nitrogen. Vascular tissues were pooled from at least 2 animals, and total RNA was extracted by use of the Trizol reagent (Takara Bio Inc. Kusatsu, Shiga, Japan). Total RNA was reverse transcribed to cDNA using the RevertAid First Strand cDNA Synthesis kit (Thermo Fisher Scientific, Waltham, MA, USA) according to the manufacturer’s protocol. The resulting cDNA was subjected to a quantitative real-time polymerase chain reaction (RT-PCR) analysis with the use of an ABI 7300 plus system and Power SYBR^®^ Green PCR Master Mix. Each sample was analyzed in triplicate and normalized to the level of GAPDH mRNA. The PCR protocol was 95 °C for 30 s, 60 °C for 30 s, and 72 °C for 30 s, for 38 cycles, followed by a final extension at 72 °C for 7 min. The primer sequences are in Supplemental Appendix A. PCR products were validated by electrophoresis on 2% agarose.

### 2.6. Western Blot Assay

Vascular samples were homogenized on ice in a lysis buffer, containing 1 × RIPA, 1 × protease and phosphatase inhibitor cocktail (MCE) using a handheld rotor-stator homogenizer (MM300; Retsch, Haan, Germany). The homogenates were centrifuged for 15 min at 12,000× *g* at 4 °C to remove debris, and aliquots of the supernatants were assayed for total protein content by the BCA method (Thermo Fisher Scientific, Waltham, MA, USA). Equal amounts of protein (10 μg per lane) were separated by 10% SDS-PAGE gels and transferred onto nitrocellulose membranes (Whatman; GE Healthcare, Chicago, IL, USA). The membranes were then incubated with the primary antibodies against EP4 (Proteintech, Wuhan, Hubei, China), PCNA (Abcam, Cambridge, UK), cyclinD1 (CST), TN-C (Abcam, Cambridge, UK), p-mTOR (CST, Danvers, MA, USA), mTOR (CST, Danvers, MA, USA), p-S6 (CST, Danvers, MA, USA), S6 (Proteintech, Wuhan, Hubei, China), and eIF5 (Snata Cruz, Dallas, TX, USA) overnight at 4 °C, followed by incubation with the appropriate secondary antibodies for 1 h at room temperature. Finally, the membranes were incubated with SuperLumia ECL Plus HRP Substrate Reagent (K22030, Abbkine, Wuhan, Hubei, China), and signals from immunoreactive bands were visualized using a Chemiluminescent Imaging System (Tanon 5200, Tanon Science & Technology, Shanghai, China).

### 2.7. Cell Culture

Rat VSMCs were isolated from the thoracic aortic arteries of male Sprague-Dawley rats (150–180 g) as described previously [30]. The cells at passages 3 to 7 were used. Each individual experiment was repeated at least 3 times with different cell preparations.

### 2.8. Scratch-Induced Wound Healing Assay

VSMCs were plated in 6-well plates. The cells were grown to 90% confluence overnight in Dulbecco’s modified Eagle’s medium supplemented with 10% fetal bovine serum and then serum-starved for 24 h. Confluent monolayers were scratched using P200 Gilson pipette tips as described [31]. Images under phase-contrast microscopy were taken at 0, 12, and 24 h. The width between the wound edges in each well at each time point was measured at fixed points by use of a standard template placed on the image. Data were expressed as wound closure rate (%) based on initial wound width.

### 2.9. Transwell Migration Assay

The chemotaxis assay was performed in transwell plates of 6.5 mm diameter with polycarbonate membrane filters containing 8 μM pores (Corning, Corning, NY, USA). The lower chamber contained PDGF-BB (10 ng/mL) as a chemoattractant. After 24 h, the cells on the upper surface were removed by gentle abrasion with a cotton bud, and the cells on the underside (invaded cells) were fixed with methanol and stained with crystal violet. Migrated cells were quantified by the average of 4 randomly chosen high-power fields of 3 independent duplicate experiments.

### 2.10. Cell Proliferation Analysis

Cell proliferation was evaluated by direct cell counting with a hematocytometer. For cell cycle analysis, VSMCs were synchronized by serum starvation for 24 h, and then they were treated with the EP4 agonists for 24 h or infected with Ad-EP4 for 36 h.

### 2.11. Immunocytochemistry

VSMCs cultured on 12 mm glass coverslips were serum-starved for 24 h and then stimulated with the EP4 agonists and antagonists for 12 or 24 h. Cells were then fixed in 4% PFA for 20 min, and permeabilized in 0.1% Triton X-100 for 10 min. Blocking of unspecific antibody activity was performed using in 2% BSA. VSMCs were then incubated with rabbit polyclonal antibody against Ki67 (Arigo, Hsinchu, Taiwan, China), or TN-C (Abcam, Cambridge, UK) diluted 1:200 in 2%BSA/PBS overnight at 4 °C. VSMCs were incubated with an Alexa Fluor^®^ 594-conjugated goat anti-rabbit IgG antibody (Molecular Probes, Invitrogen, Carlsbad, CA, USA) diluted 1:500 in DAPI/PBS for 1 h at room temperature. The cells were examined using a Leica TCS SP8 (Leica Microsystems, Wetzlar, Germany) laser scanning fluorescence confocal microscope.

### 2.12. MTS Assay

VSMCs were plated on 96 well plates at 5000 cells in 100 μL of DMEM per well and incubated in the presence of PDGF-BB (10 ng/mL) for 24 h. Then, 20 μL of a mixture of tetrazolium compound and phenazine methosulfate was added for an MTS assay, and the absorbance was detected at 490 nm.

### 2.13. Gene Silencing by siRNA

The small interfering RNA (siRNA) sequences targeting rat EP4 was as follows: 5′-GCUGAGAACUUUGCGAAUUTT-3′ (sense), 5′-AUUCGCAAAGUUCUCAGCTT-3′ (antisense). The siRNA sequences targeting rat TN-C was as follows: siRNA-1, 5′-CCACUGAGUACGAAAUUGATT-3′ (sense), 5′-UCAAUUUCGUACUCAGUGGTT-3′ (antisense); siRNA-2, 5′-GACAGUGUGUUUGCAACGATT-3′ (sense), 5′-UCGUUGCAAACACACUGUCTT-3′ (antisense); siRNA-3, 5′-CAGAUGAUCUGGCCUAUAATT-3′ (sense), 5′-UUAUAGGCCAGAUCAUCUGTT-3′ (antisense). The siRNA with scrambled sequence was used as negative control (Scr siRNA): sequence as follow: 5′-UUCUCCGAACGUGUCACGUTT-3′ (sense), 5′-ACGUGACACGUUCGGAGAATT-3′ (antisense). The siRNAs (2.5 μg) were transfected into the cells using Lipofectamine 3000 according to the manufacturer’s protocol.

### 2.14. Statistics

All data are presented as mean ± SEM. Statistical analyses were performed using GraphPad Prism 8 software. Comparisons between 2 groups were tested by 2-tailed Student’s t test. Comparisons among multiple groups were made by ANOVA. *p* < 0.05 was considered statistically significant.

## 3. Results

### 3.1. Vascular EP4 Expression Is Increased in Injured Arteries

To investigate the role of EP4 in vascular injury, we determined whether the expression of EP4 was altered in injured arteries. Wire-injured femoral artery model and carotid artery ligation model were generated in mice. Quantitative RT-PCR and Western blot analysis showed that the mRNA and protein levels of EP4 were dramatically elevated in the femoral arteries at 28 days after wire injury (Figure 1A–C). We also found a significant increase in EP4 expression at both mRNA and protein levels in the carotid arteries after ligation for 28 days (Figure 1D–F). These data suggest that elevated EP4 expression may play an important role in the development of intimal hyperplasia.

### 3.2. Specific Deletion of EP4 in VSMCs Ameliorates Neointimal Hyperplasia

To determine the contribution of the EP4 receptor to vascular neointimal formation in vivo, we generated a mouse with VSMC-specific EP4 gene knockout (VSMC-EP4^−/−^) by crossing the EP4 flox/flox mice (EP4^f/f^) with the SMMHC-Cre mice as previously reported (Appendix A) [19]. The mice were subjected to femoral artery injury by wire for 28 days. Hematoxylin and eosin (H&E) staining showed an obvious reduction in the neointimal area and intima-to-media ratio (I/M ratio) of the VSMC-EP4^−/−^ mice compared with the EP4^f/f^ mice (Figure 2A), suggesting that VSMC-specific ablation of EP4 attenuates neointimal hyperplasia in wire-injured femoral artery. To further confirm this finding, we also examined the role of VSMC EP4 in a murine carotid artery ligation model. Consistently, the VSMC-EP4^−/−^ mice exhibited a significant reduction in the intima area as well as the I/M ratio compared with the EP4^f/f^ mice (Figure 2B). However, in both models the medial VSMC layer areas were similar between two genotypes (Figure 2A,B). These results demonstrate that VSMC EP4 promotes arterial neointimal formation after vascular injury.

### 3.3. Overexpression of Human EP4 in VSMCs Exacerbates Neointimal Formation in Mice

To investigate the effect of EP4 overexpression in VSMC on neointima formation, a transgenic mouse with VSMC-specific human EP4 gene expression (VSMC-hEP4 Tg) was created, as previously reported (Appendix A) [28], and used for generating neointimal hyperplasia models. As expected, H&E staining of the femoral arteries and carotid arteries revealed that specific overexpression of human EP4 in VSMCs aggravated neointimal hyperplasia at 28 days after vascular injury, as evidenced by semi-quantification of the neointima area and I/M ratio (Figure 2C,D), and no significance difference in the media area was observed between the WT and VSMC-hEP4-Tg mice (Figure 2C,D). Additionally, arterial infection of an adenovirus carrying GFP (Ad-GFP) or EP4 (Ad-EP4) was performed in the mice with femoral artery injury for 14 days. The results show that Ad-EP4 infection markedly increased arterial EP4 protein expression, as assessed by immunohistochemical staining (Appendix A), and accelerated neointimal formation in wire-injured femoral arteries, with no change in the media areas (Appendix A). Taken together, these data strongly suggest that VSMC-specific overexpression of human EP4 accelerates vascular neointimal formation.

### 3.4. EP4 Facilitates VSMC Proliferation In Vivo and In Vitro

To examine the response of VSMCs to vascular injury, we quantified the percentage of proliferating cell nuclear antigen positive cells (PCNA^+^) within the neointima on day 28 after wire-mediated injury or artery ligation. We found that the numbers of the neointimal PCNA^+^ cells were significantly reduced in the VSMC-EP4^−/−^ mice compared with that in the EP4^f/f^ mice (Appendix A). On the contrary, the numbers of the PCNA^+^ cells were markedly increased in the neointima of the VSMC-hEP4 Tg mice compare with the WT mice (Appendix A). Similarly, arterial infection of Ad-EP4 dramatically increased the percentage of the PCNA^+^ cells by ~2-fold as compared to the Ad-GFP-infected arteries (Appendix A). These results further suggest that EP4 in VSMCs promotes VSMC proliferation induced by wire injury or artery ligation.

To confirm this finding, we determined the effect of selective activation or overexpression of the EP4 receptor on cell proliferation of cultured rat VSMCs. We found that treatment of rat VSMCs with the EP4 inhibitors, MF498 and L161982, for 24 h dose-dependently inhibited fetal bovine serum (FBS)- and platelet-derived growth factor BB (PDGF-BB)-induced VSMC proliferation, as assessed by cell number counting and the MTS assay (Figure 3A,B). In support, inhibition of EP4 markedly reduced FBS- and PDGF-induced Ki67 expression in VSMCs (Figure 3C,D). Conversely, the EP4 agonists PGE1-OH and CAY10580 significantly increased the numbers of Ki67 positive cells (Appendix A). Furthermore, Western blot assay revealed that both MF498 and L161982 significantly mitigated PDGF-induced cyclin D1 and PCNA expression (Figure 3E–H). In cultured VSMCs, treatment with PGE1-OH or infection of Ad-EP4 also significantly upregulated the mRNA levels of genes related to cell proliferation including Ki67, Foxm1, Ccna2, Ccnb1, Ccnd1, Ccnd2, Ccne1, and CDK2 (Appendix A). Collectively, these findings demonstrate that EP4 promotes VSMC proliferation, contributing to enhanced neointimal growth and restenosis in response to vascular injury.

### 3.5. Pharmacological Activation of EP4 Promotes VSMCs Migration In Vitro

In addition to the proliferation, the migration of VSMCs is also a critical process underpinning the development of neointima hyperplasia [32]. Therefore, we performed wound closure and transwell assays in cultured VSMCs to determine the effect of EP4 on cell migration. As shown in wound closure assays, both MF498 and L161982 effectively mitigated the PDGF-BB-induced migration (Figure 4A,B). The transwell analyses also showed that inhibition of EP4 by MF498 and L161982 largely abrogated pro-migratory effect of PDGF-BB (Figure 4C,D). Similarly, siRNA-mediated knockdown of EP4 also blocked the PDGF-BB-induced migration of VSMCs (Appendix A). In contrast, the EP4 agonist PGE1-OH greatly promoted cell migration, as determined by both wound closure and transwell assays (Figure 4E–H). Altogether, these data indicate that activation of EP4 could promote the VSMC migration.

### 3.6. EP4 Induces Proteoglycan Tenascin Expression in Injured Arteries and in Cultured VSMCs

Previous studies have reported that tenascin C (TN-C) directly contributes to neointimal hyperplasia by promoting neointimal cell migration and proliferation after aortotomy in mice [27]. More interestingly, in a wire-injured femoral artery model, Wang et al. reported that deprivation of mPGES-1 in mice mitigates vascular neointimal hyperplasia by reducing TN-C expression [9]. In our study, immunohistochemical staining of the injured femoral arteries and ligated carotid arteries with an antibody against TN-C showed that TN-C protein expression was elevated in the medial and neointimal layers following vascular injury, which was reversed by specific deletion of EP4 in VSMCs (Figure 5A,B). Conversely, injury-stimulated induction of TN-C expression was dramatically higher in the VSMC-hEP4 Tg mice compared with the WT mice (Figure 5C,D). These findings strongly suggest that VSMC EP4 levels are correlated with the TN-C protein expression in injured arteries. Additionally, in vitro studies using rat VSMCs revealed that administration of the EP4 agonists (PGE1-OH and CAY10580) remarkably upregulated TN-C protein expression as assessed by Western blot and immunofluorescence staining (Figure 5E–H). Collectively, these findings demonstrate that EP4 expression is correlated with TN-C expression in injured vessels and EP4 activation induces TN-C expression in cultured VSMCs.

### 3.7. EP4 Upregulates TN-C Protein Expression at Translational Level

To determine the underlying mechanism by which EP4 promotes TN-C protein expression, we examined the TN-C mRNA expression in VSMCs treated with the EP4 agonists by using qPCR technique. We found that EP4 activation by PGE1-OH or CAY10580 had little effect on TN-C mRNA expression, suggesting a post-transcriptional mechanism underpinning the EP4-elicited TN-C protein expression (Figure 6A). As expected, pretreatment of VSMCs with cycloheximide (CHX), a translational elongation inhibitor, remarkably blocked the up-regulation of TN-C protein expression induced by the EP4 agonists (Figure 6B,C). In addition, administration of PGE1-OH or CAY10580 had no obvious effect on the rate of TN-C protein turnover (Figure 6D,E), suggesting that EP4 is not involved in the post-translational modification of TN-C protein. These data indicate that EP4 promotes TN-C protein expression at translational level.

### 3.8. EP4 Upregulates TN-C Protein Expression via the PKA/mTOR/rpS6 Pathway

It has been well documented that ribosomal protein activity is important in the translational process. Ribosomal protein S6 (rpS6 or S6) is the major substrate of protein kinases in eukaryote ribosomes and its phosphorylation at ser-235 and ser-236 facilitates the assembly of the preinitiation complex [33]. We found both PGE1-OH and CAY10580 significantly up-regulated S6 phosphorylation (Figure 7A–D), suggesting that activation of EP4 may enhance TN-C translation by increasing ribosome activity. To investigate how S6 is phosphorylated by EP4, we examined the activity of mammalian target of rapamycin (mTOR), which is the upstream kinase of S6 capable of promoting the S6 phosphorylation. The results show that the EP4 agonists remarkably increased the phosphorylation of mTOR as well as S6, which was completely blocked by the administration of rapamycin, an inhibitor of mTORC1 (Figure 7A,B). Therefore, these findings indicate that EP4 increases the phosphorylation of S6 in an mTORC1-p70S6K dependent manner.

Since the cAMP/PKA pathway mediates most of important biological functions of EP4 [34], we then tested the hypothesis that EP4-elicited TN-C protein expression is dependent on the cAMP/PKA cascade. We found that pretreatment of VSMCs with H89, a selective PKA inhibitor, completely blocked the phosphorylation of mTOR and S6 induced by the EP4 agonists (Figure 7C,D). Moreover, pretreatment with H89 almost completely abolished the EP4-activation-induced TN-C protein expression (Figure 7E,F). Similarly, H89 treatment suppressed Ad-EP4-promoted TN-C protein expression (Appendix A). These results support the possibility that EP4-elicited TN-C protein expression is dependent on the PKA activity. Together, our findings demonstrate that EP4 increases TN-C protein expression by the PKA/mTORC1/rpS6 pathway.

### 3.9. TN-C Mediates EP4-Elicited VSMC Proliferation and Migration

To investigate whether TN-C is responsible for EP4-induced VSMC proliferation and migration, in vitro studies using TN-C siRNA were performed to silence endogenous TN-C protein expression in cultured VSMCs. Western blot analysis showed that the protein levels of TN-C were markedly reduced in VSMCs transfected with the TN-C siRNA-2, which was then used in the subsequential studies (Figure 8A). The proliferation of VSMCs induced by the EP4 agonists PGE1-OH and CAY10580 was significantly suppressed by siRNA-mediated TN-C knockdown (Figure 8B,C and Appendix A). Consistently, the pro-migratory effect of PGE1-OH and CAY10580 was eliminated by TN-C gene silencing (Figure 8D,E and Appendix A). Collectively, these results demonstrate that EP4-elicited VSMC proliferation and migration relies on the TN-C expression.

## 4. Discussion

A large body of evidence demonstrates that COX-2-derived PGE2 and its G protein-coupled receptors play a critical role in maintaining vascular homeostasis and dysfunction of the PGE2-EPs axis contributes to the pathogenesis of many cardiovascular diseases including hypertension and vascular remodeling such as patent ductus arteriosus and aortic dissection [11,19,21,22]. It has also been reported that the PGE2 receptor EP2 and EP3 exert an opposite effect on the pathogenesis of neointimal formation via regulating VSMC proliferation and/or migration [11,18]. In the present study, by generating wire-injured femoral artery and carotid artery ligation models of vascular restenosis using VSMC-specific knockout (VSMC-EP4^−/−^) or overexpression (VSMC-hEP4 Tg) mice, we demonstrated that VSMC EP4 promotes neointimal formation by increasing VSMC proliferation and migration. We further uncovered a novel mechanism by which EP4 enhances VSMC proliferation and migration via up-regulating TN-C protein expression at translational level through the PKA/mTORC1/rpS6 pathway.

Proliferation and migration of VSMCs in the arterial wall are two critical pathological processes in the development of neointimal hyperplasia underlying obstructive vascular diseases, such as atherosclerosis and restenosis after percutaneous transluminal coronary angioplasty [35]. We found that VSMC-specific deletion of EP4 remarkably suppressed, while VSMC-selective overexpression of EP4 exaggerated, the PCNA positive cells in the vascular neointima, suggesting that VSMC EP4 may promote neointimal hyperplasia via enhancing VSMC proliferation. This possibility was further supported by our in vitro studies using pharmacological approaches. Moreover, wound healing and transwell assays revealed that activation of EP4 accelerated, while inactivation of EP4 suppressed, the PDGF-BB-induced VSMC migration. Therefore, VSMC EP4 may accelerate vascular neointimal hyperplasia via promoting VSMC proliferation and migration.

It has been well documented that the production of ECM is critical for the formation of vascular neointima. Among many ECM proteins, TN-C, an extracellular matrix glycoprotein, is up-regulated during neointimal hyperplasia and closely associated with the phenotypic change and increased proliferation and migration of VSMCs. Yamamoto et al. reported that in a murine aortotomy model TN-C deficiency almost completely abolished neointimal formation, with a significant reduce in proliferating cell nuclear antigen (PCNA) expression, suggesting that TN-C is a crucial molecule in the pathogenesis of neointimal hyperplasia [27,36]. In a rat model of pulmonary vascular disease, TN-C was colocalized with proliferating VSMCs in pulmonary arterials and promoted epidermal growth-factor- and basic fibroblast growth-factor-induced cell growth in cultured VSMCs [37]. Similarly, in balloon-catheter-injured rat carotid arteries, TN-C appeared in the neointima formed by proliferating VSMCs and was increased in cultured VSMCs with phenotypic change [25]. These findings demonstrate that TN-C production is induced concomitantly with the change in VSMC phenotype and promotes VSMC proliferation. In addition, Wang et al. have reported that the deletion of mPGES-1 attenuates the induction of TN-C in response to vascular injury, suggesting mPGES-1-derived PGE2 may increase TN-C expression in VSMCs. They also found that knockdown of TN-C in cultured VSMCs impairs cell migration [9]. These results indicate that TN-C increases the migration of proliferating VSMCs to the tunica intima to facilitate the neointimal formation. Therefore, TN-C is essential in the development of vascular neointimal hyperplasia.

In the present study, we provide direct evidence that EP4 increases VSMC TN-C protein expression in vivo and in vitro. We found that genetic ablation of EP4 in VSMCs significantly reduced, while VSMC-specific overexpression of human EP4 strongly promoted, TN-C protein expression in the neointima of injured arteries. Two structurally distinct EP4 agonists, i.e., PGE1-OH and CAY10580, markedly up-regulated TN-C protein levels in cultured VSMCs. EP4-elicited TN-C protein expression appears to be at post-transcriptional level, since neither PGE1-OH nor CAY10580 altered the TN-C mRNA expression. In addition, EP4 activation did not change the half-life of TN-C protein, indicating that EP4 is not involved in the degradation of TN-C protein. Surprisingly, we found that pharmacological activation of EP4 remarkably induced the phosphorylation of mTOR (Ser2448) and its downstream effector rpS6 (Ser235/236) and blocking the protein synthesis process (elongation) in the ribosome eliminated the TN-C protein up-regulation induced by the EP4 agonists. Altogether, given an essential role of TN-C in the neointimal formation, these findings demonstrate that EP4 may promote neointimal formation by increasing TN-C expression in VSMCs at the translational level.

Ribosomal protein S6 is an inducible phosphorylatable protein and a downstream effector of the mTORC1 and p70S6K, which is essential for ribosome function [33] and important for cell proliferation, growth, and migration [38]. We found that inhibition of mTORC1 by rapamycin completely blocked the increase of S6 phosphorylation at Ser235/236 upon EP4 activation, indicating that EP4 promotes S6 phosphorylation at Ser235/236 in an mTORC1-dependent manner. It has been well known that EP4 exerts its biological function mainly through the cAMP/PKA pathway [39] and PKA may directly or indirectly phosphorylate S6 at Ser235/236 [40,41]. To characterize the mechanism by which EP4 enhances S6 phosphorylation, we blocked PKA by H89 and found that inhibition of PKA completely suppressed mTOR phosphorylation (Ser2448) as well as S6 phosphorylation (Ser235/236). Together, these data clearly indicate that EP4 indirectly promotes S6 phosphorylation (Ser235/236) via the PKA/mTORC1 pathway rather than through PKA directly. Since mTOR as a serine/threonine protein kinase plays an important role in cell proliferation and migration [42,43], its inhibitor rapamycin has been used in clinic to prevent restenosis [44,45,46,47]. The present study provides a novel insight into the mechanism by which rapamycin attenuates vascular neointimal hyperplasia through downregulation of TN-C. Furthermore, our findings that EP4 promotes neointimal formation via the PKA-mTORC1-rpS6 pathway may have potential clinical implication. Blocking EP4 activation or PKA activity may represent an attractive therapeutic option to treat vascular restenosis.

It has been previously reported that TN-C is regulated by virous factors such as cytokines/chemokines, growth factors, and inflammatory mediators through multiple signaling pathways including the transforming growth factor (TGF)β/Smads, toll-like receptor (TLR)4/nuclear factor-κB (NF-κB), PDGF/phosphoinositide 3-kinase/Akt, and PDGF/MAPK cascades [48]. As a lipid proinflammatory mediator, PGE2 was reported to regulate TN-C expression in VSMCs [9]. Here, we provide a novel underlying mechanism that EP4 controls the TN-C protein level at post-transcriptional level.

In addition, down-stream of PGE2 and cAMP, exchange protein activated by cAMP (EPAC) has also been implicated in neointimal hyperplasia development [49]. However, the effect of EPAC on neointimal hyperplasia is paradoxical, while EPAC activation may act as a modulator of PKA signaling [50]. Our study observed that the expression of TN-C was almost completely blocked by pharmacological inhibition of PKA by H89, indicating a predominant effect of the cAMP-PKA pathway elicited by EP4. However, it does not exclude the possibility that EP4 may also play a role in the proliferation and migration of VSMCs through the cAMP-EPAC pathway.

It has been previously reported that deletion of endothelial EP4 exacerbates neointimal hyperplasia after wire injury [20]. EP4 can promote endothelial cell (ECs) migration and angiogenesis [51,52], which appears to be beneficial to repair vascular damage. Therefore, deficiency of endothelial EP4 aggravates neointimal formation. Nevertheless, the key cellular component of the neointima is migratory and proliferative VSMCs rather than endothelial cells [53]. In the present study, we found that EP4 promoted the proliferation and migration of VSMCs, leading to the acceleration of neointimal formation. Therefore, although EP4 increased the proliferation of both ECs and VSMCs, it exerted the opposite effects on neointimal hyperplasia possibly due to distinct functions of different cell types in the artery.

However, this study has some limitations. First, the study was performed only in male mice. Gender differences may lead to different observations. Second, we only used mice overexpressing human EP4 to study the role of EP4 in humans. EP4 may has distinct role in different species. Third, PGE_2_ acts as a proinflammatory mediator, the role of EP4 receptor may be different under conditions of obesity or other diseases.

## 5. Conclusions

In summary, we found that specific deletion of EP4 in VSMCs significantly attenuated, while selective overexpression of human EP4 gene in VSMCs markedly aggravated, neointimal hyperplasia. We further showed that activation of EP4 in VSMCs up-regulated TN-C protein expression at translational level via the PKA/mTOR/rpS6 pathway. Our findings demonstrate that the VSMC EP4 is a critical player in the pathogenesis of neointimal formation and may represent a promising target for the therapeutic intervention of vascular restenosis.

## Figures and Tables

**Figure 1 cells-11-02720-f001:**
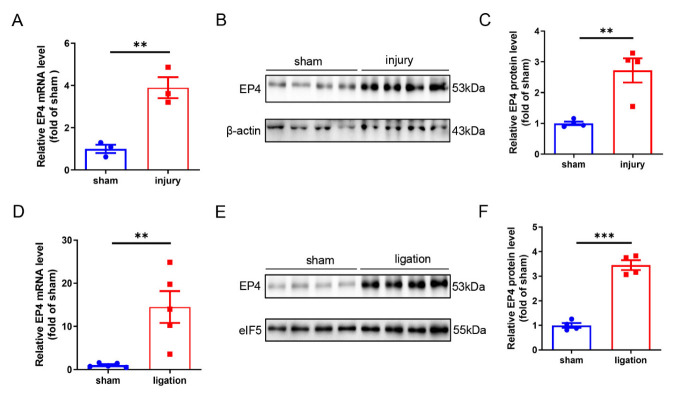
EP4 expression was increased after vascular injury. (**A**). qRT-PCR analysis of the EP4 mRNA expression in mouse femoral artery after 28 days of wire injury. *n* = 3 per group. (**B**,**C**). Western blot analysis of the EP4 protein levels in mouse femoral artery after 28 days of wire injury. *n* = 4 per group. (**D**). qRT-PCR analysis of the EP4 mRNA expression in mouse carotid artery after 28 days of ligation. *n* = 5 per group. (**E**,**F**). Western blot analysis of the EP4 protein expression in mouse carotid artery after 28 days of ligation. Quantitative result was analyzed (**F**). *n* = 4 per group. Data were presented as mean ± SEM, ** *p* < 0.01, *** *p* < 0.001.

**Figure 2 cells-11-02720-f002:**
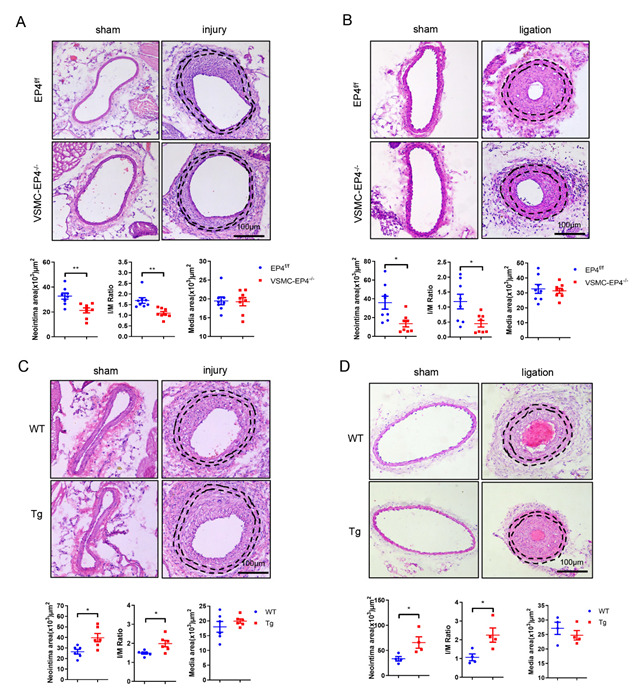
Effect of VSMC-specific deletion or overexpression of EP4 on neointimal formation. (**A**). Representative H&E staining of femoral arteries in the EP4^f/f^ and VSMC-EP4^−/−^ mice with femoral artery wire injury for 28 days. The neointima area, intima-to-media (I/M) ratio and media area were analyzed. *n* = 8 per group. (**B**). Representative H&E staining of carotid arteries in the EP4^f/f^ and VSMC-EP4^−/−^ mice with carotid artery ligation for 28 days. The neointima area, intima-to-media (I/M) ratio and media area were analyzed. *n* = 8 per group. (**C**). Representative H&E staining of femoral arteries in WT and the VSMC-hEP4 Tg mice with femoral artery wire injury for 28 days. The neointima area, intima-to-media (I/M) ratio and media area were analyzed. *n* = 6 per group. (**D**). Representative H&E staining of carotid arteries in WT and the VSMC-hEP4 Tg mice with carotid artery ligation for 28 days. The neointima area, intima-to-media (I/M) ratio and media area were analyzed. *n* = 4 per group. Black dotted line indicates internal and external elastic lamina. Between the black dotted line is the media area. Scale bars = 100 μM. Data are presented as mean ± SEM. * *p* < 0.05, ** *p* < 0.01.

**Figure 3 cells-11-02720-f003:**
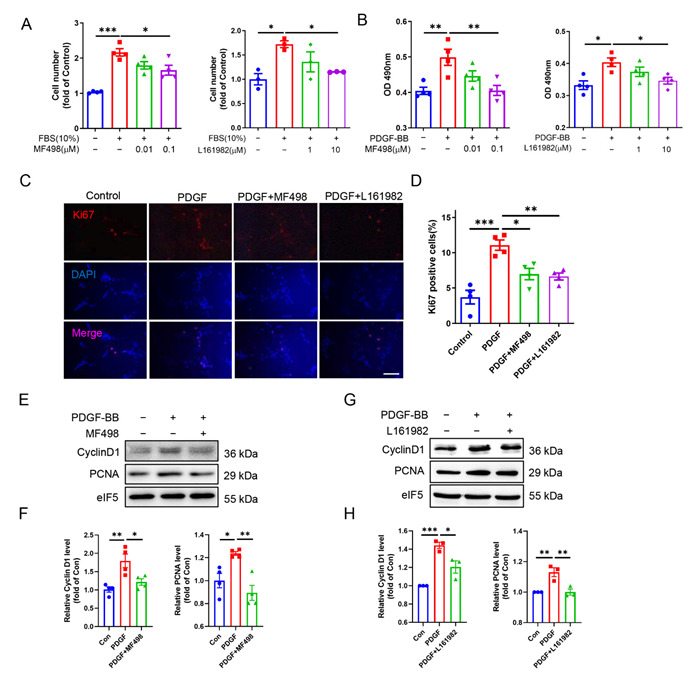
EP4 facilitates VSMC proliferation in vitro. (**A**). Rat VSMCs were pretreated with MF498 (0.01 μM, 0.1 μM) or L161982 (1 μM, 10 μM) for 30 min, and then FBS (10%) was added to induce cell proliferation for 24 h. The cell numbers were counted at the end point of experiment. *n* = 3–4. (**B**). Rat VSMCs were pretreated with MF498 (0.01 μM, 0.1 μM) or L161982 (1 μM, 10 μM) for 30 min, and the cells were then treated with PDGF-BB (10 ng/mL) for 24 h. The cell viability was measured by MTS assay. *n* = 4. (**C**,**D**). Rat VSMCs were pretreated with MF498 (0.1 μM) or L161982 (10 μM) for 30 min, and then PDGF-BB (10 ng/mL) was added to the cells for 24 h. Immunofluorescence analysis of Ki67 was performed and the quantification of Ki67 positive cells was shown in (**D**). Scale bars: 100 μM. *n* = 4. (**E**,**F**). Rat VSMCs were pretreated with MF498 (0.1 μM) for 30 min and then treated with PDGF-BB (10 ng/mL) for 12 h. Western blot analysis was performed to check Cyclin D1 and PCNA protein levels (**E**). Quantification was performed using the Image J software (**F**). *n* = 4. (**G**,**H**). Rat VSMCs were pretreated with L161982 (10 μM) for 30 min and then treated with PDGF-BB (10 ng/mL) for 12 h. Western blot analysis was performed to measure Cyclin D1 and PCNA protein level (**G**). Quantification was performed using the Image J software (**H**). *n* = 3. Data are presented as mean ± SEM. * *p* < 0.05, ** *p* < 0.01, *** *p* < 0.001.

**Figure 4 cells-11-02720-f004:**
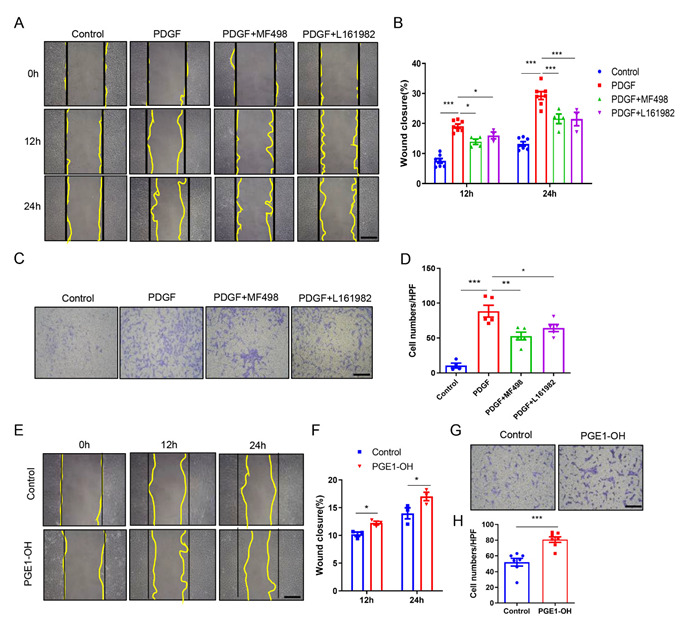
EP4 promotes VSMC migration in vitro. (**A**,**B**). Rat VSMCs were scratched to perform wound healing assay. The cells were pretreated with MF498 (0.1 μM) or L161982 (10 μM) for 30 min and PDGF-BB (10 ng/mL) was then added to induce migration. Images were taken under light microscope after 12 and 24 h (**A**) and the closure rates were calculated (**B**). Scale bars: 100 μM. *n* = 4–7. (**C**,**D**). Rat VSMCs were cultured on the upper surface of a transwell plate and were pretreated with MF498 (0.1 μM) or L161982 (10 μM) for 30 min. PDGF-BB (10 ng/mL) was then added in the lower chamber for 24 h. The cells on the underside were stained with crystal violet (**C**). Migrated cells were quantified by the average of 4 random fields (**D**). *n* = 4–5. (**E**,**F**). Rat VSMCs were scratched and treated with PGE1-OH (0.1 μM). Images were taken under light microscope after 12 and 24 h (**E**) and the closure rates were calculated (**F**). Scale bars: 100 μM. *n* = 3–7. (**G**,**H**). Rat VSMCs were cultured on the upper surface of a transwell plate and treated with PGE1-OH (0.1 μM) for 24 h. The cells on the underside were stained with crystal violet (**G**). Migrated cells were quantified by the average of 4 random fields (**H**). *n* = 7. Data are presented as mean ± SEM. * *p* < 0.05, ** *p* < 0.01, *** *p* < 0.001.

**Figure 5 cells-11-02720-f005:**
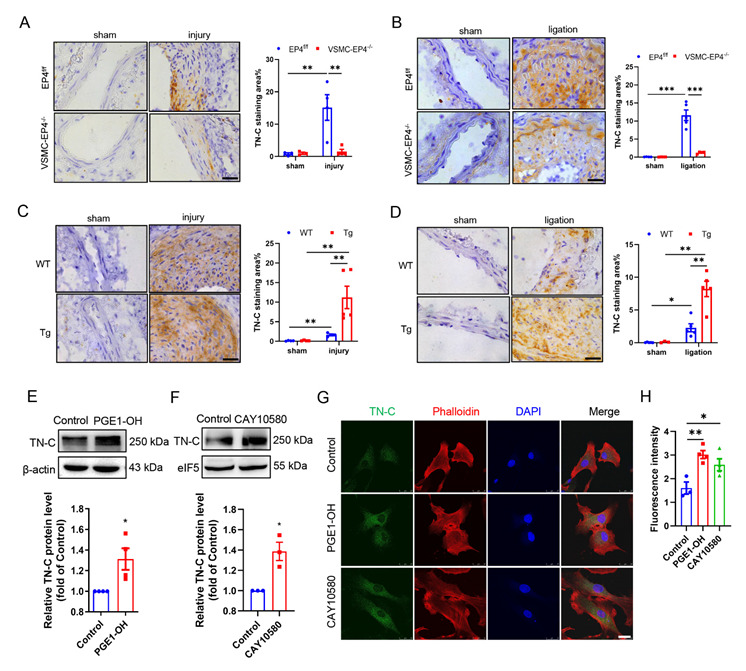
EP4 upregulates TN-C expression in injured artery. (**A**,**B**). Representative immunohistochemical analysis of TN-C after femoral artery wire injury (**A**) and carotid artery ligation (**B**) in the EP4^f/f^ and VSMC-EP4^−/−^ mice. Quantification was performed using the Image J. Scale bars: 25 μM. *n* = 4–5. (**C**,**D**). Representative immunohistochemical analysis of TN-C after femoral artery wire injury (**C**) and carotid artery ligation (**D**) in WT and the VSMC-hEP4-Tg mice. Quantitative analysis was performed using the Image J. Scale bars: 25 μM. *n* = 4–5. (**E**,**F**). Rat VSMCs were treated with PGE1-OH (0.1 μM) or CAY10580 (1 μM) for 24 h. TN-C protein levels were examined by Western blot. *n* = 3–4. (**G**,**H**). Rat VSMCs were treated with PGE1-OH (0.1 μM) or CAY10580 (1 μM) for 24 h. Immunofluorescence analysis of TN-C was performed. The quantitative results were showed in (**H**). Scale bars: 25 μM. *n* = 3–4. Data are presented as mean ± SEM. * *p* < 0.05, ** *p* < 0.01, *** *p* < 0.001.

**Figure 6 cells-11-02720-f006:**
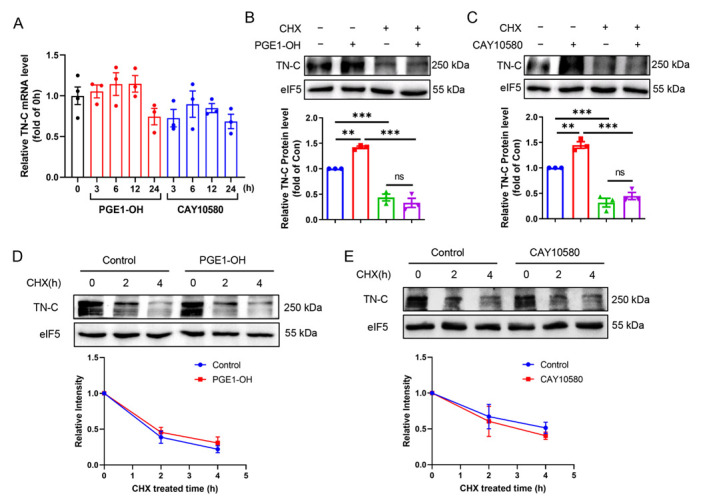
EP4 promotes TN-C protein expression at translational level. (**A**). Rat VSMCs were treated with PGE1-OH (0.1 μM) or CAY10580 (1 μM) for different periods of time and the TN-C mRNA expression was measured by qPCR. *n* = 3–4. (**B**,**C**). Rat VSMCs were pretreated with CHX (10 μM) for 1 h and then treated with PGE1-OH (0.1 μM) (**B**) or CAY10580 (1 μM) (**C**) for 24 h. TN-C protein expression was examined by Western blot. Quantification was performed using the Image J. *n* = 3. (**D**,**E**). Rat VSMCs were pretreated with PGE1-OH (0.1 μM) (**D**) or CAY10580 (1 μM) (**E**) for 1 h and then treated with CHX (10 μM) for indicated time. TN-C protein degradation was examined by Western blot. Quantification was performed using the Image J. *n* = 3. Data are presented as mean ± SEM. ** *p* < 0.01, *** *p* < 0.001.

**Figure 7 cells-11-02720-f007:**
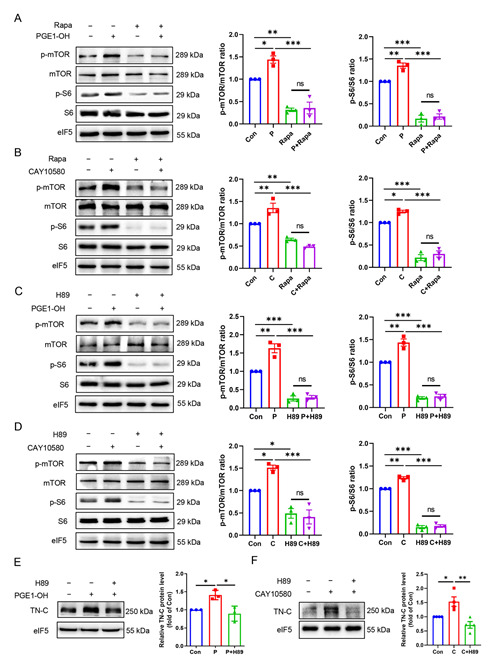
EP4 increases TN-C protein expression via the PKA/mTORC1/S6 pathway. (**A**,**B**). Rat VSMCs were pretreated with rapamycin (Rapa, 10 nM) for 0.5 h and then treated with PGE1-OH (0.1 μM) (**A**) or CAY10580 (1 μM) (**B**) for 15 min. Phosphorylated mTOR and S6, and total mTOR and S6 levels were examined by Western blot. Quantification was performed using the Image J. *n* = 3. (**C**,**D**). Rat VSMCs were pretreated with H89 (10 μM) for 0.5 h and then treated with PGE1-OH (0.1 μM) (**C**) or CAY10580 (1 μM) (**D**) for 15 min. Phosphorylated mTOR and S6, and total mTOR and S6 levels were examined by Western blot. Quantification was performed using the Image J. *n* = 3. (**E**,**F**). Rat VSMCs were pretreated with H89 (10 μM) for 0.5 h and then treated with PGE1-OH (0.1 μM) (**E**) or CAY10580 (1 μM) (**F**) for 24 h. TN-C protein levels were examined by Western blot. Quantification was performed using the Image J. *n* = 3. Data are presented as mean ± SEM. * *p* < 0.05, ** *p* < 0.01, *** *p* < 0.001.

**Figure 8 cells-11-02720-f008:**
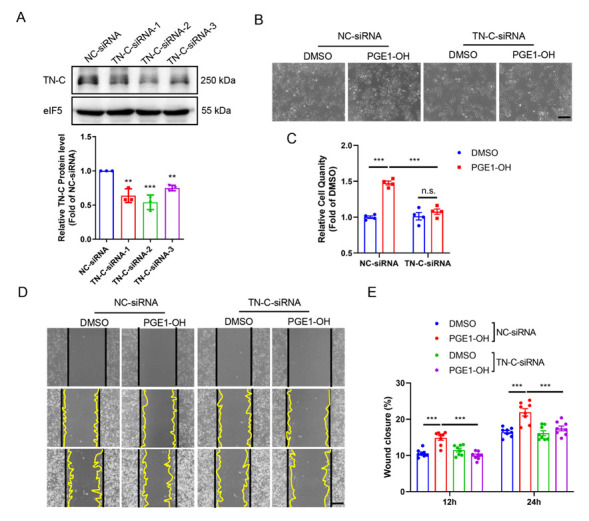
TN-C mediates EP4-elicited VSMC proliferation and migration. (**A**). Rat VSMCs were transfected with 3 different TN-C siRNA (siRNA-1, siRNA-2 and siRNA-3) for 48 h. The TN-C protein expression was examined by Western blot. Quantification was performed using the Image J. *n* = 3. (**B**,**C**). Rat VSMCs were transfected with the TN-C siRNA-2 for 24 h and then treated with PGE1-OH (0.1 μM) for 24 h. The cell numbers were counted at the end of experiment and relative cell quantity was presented in (**C**). *n* = 4. (**D**,**E**). Rat VSMCs were transfected with the TN-C siRNA-2 for 24 h and then scratched. PGE1-OH (0.1 μM) was treated and the images were taken under light microscope at the indicated time point (12 and 24 h) (**D**) and the closure rates were calculated (**E**). *n* = 4–7. Scale bars: 100 μM. Data are presented as mean ± SEM. ** *p* < 0.01, *** *p* < 0.001.

## Data Availability

The data presented in this study are available in [insert article or Appendix A here].

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
