# Peer review of "The Prostaglandin E2 Receptor EP4 Promotes Vascular Neointimal Hyperplasia through Translational Control of Tenascin C via the cAMP/PKA/mTORC1/rpS6 Pathway"

_cells, 2022, doi:10.3390/cells11172720_

Round 1
Reviewer 1 Report
GENERAL COMMENTS
The topic is interesting as it provides important insight into the signaling pathway involved in VSMC neointimal hyperplasia and proliferation. In this sense, the manuscript addresses a worthwhile topic. The authors have performed a wealth of experiments. Some suggestions are provided to focus on some additional aspects.
The manuscript may benefit from considering the following aspects:
At the end of the Introduction, page 2, line 84: formulate the specific working hypothesis – what did you expect and why.
Page 6, line 263: replace “reduce” by “reduction”.
Discussion
Page 16, lines 481-484: delete these sentences.
TnC, an extracellular matrix protein, is transiently expressed during tissue injury and plays a role in fibrogenesis and tumorigenesis in other tissues (for example, ref Benbow JH, Thompson KJ, Cope HL, Brandon-Warner E, Culberson CR, Bossi KL, Li T, Russo MW, Gersin KS, McKillop IH, deLemos AS, Schrum LW. Diet-Induced Obesity Enhances Progression of Hepatocellular Carcinoma through Tenascin-C/Toll-Like Receptor 4 Signaling. Am J Pathol. 2016 Jan;186(1):145-58). Enhanced Toll-like receptor 4 signaling activated by TnC, reportedly promotes an increased inflammatory response, that has been shown in hepatocyte transformation, and migration. It may be reasonable to think that other signalling pathways apart from the cAMP/PKA/mTORC1/rpS6 might be also activated by TnC in VSMC.
In the Discussion, it would be interesting to provide some more translational perspective. Besides its role in body weight control leptin also acts as a vasoactive hormone (ref Trovati M, Doronzo G, Barale C, Vaccheris C, Russo I, Cavalot F. Leptin and vascular smooth muscle cells. Curr Pharm Des. 2014;20(4):625-34). The expression of functional leptin receptors has been detected in VSMCs. Moreover, leptin modifies angiotensin II-induced vascular responses (ref Fortuño A, Rodríguez A, Gómez-Ambrosi J, Muñiz P, Salvador J, Díez J, Frühbeck G. Leptin inhibits angiotensin II-induced intracellular calcium increase and vasoconstriction in the rat aorta. Endocrinology. 2002 Sep;143(9):3555-60). These additional intracellular mechanisms can be mentioned in the Discussion to broaden the implications of the findings.
At the end of the Discussion, indicate limitations of the study; e.g. the work was performed only in males, the signalling may be altered in pathophysiological circumstances like obesity.
Author Response
1. At the end of the Introduction, page 2, line 84: formulate the specific working hypothesis – what did you expect and why.
Response: Thanks for your helpful suggestion. We have formulated a specific working hypothesis in the section of “Introduction”.
2. Page 6, line 263: replace “reduce” by “reduction”.
Response: Many thanks for your suggestion. We have replaced “reduce” by “reduction”.
3. Page 16, lines 481-484: delete these sentences.
Response: Thank you for your kind suggestion. We have deleted the sentences.
4. TnC, an extracellular matrix protein, is transiently expressed during tissue injury and plays a role in fibrogenesis and tumorigenesis in other tissues (for example, ref Benbow JH, Thompson KJ, Cope HL, Brandon-Warner E, Culberson CR, Bossi KL, Li T, Russo MW, Gersin KS, McKillop IH, deLemos AS, Schrum LW. Diet-Induced Obesity Enhances Progression of Hepatocellular Carcinoma through Tenascin-C/Toll-Like Receptor 4 Signaling. Am J Pathol. 2016 Jan;186(1):145-58). Enhanced Toll-like receptor 4 signaling activated by TnC, reportedly promotes an increased inflammatory response, that has been shown in hepatocyte transformation, and migration. It may be reasonable to think that other signalling pathways apart from the cAMP/PKA/mTORC1/rpS6 might be also activated by TnC in VSMC.
Response: We thank the reviewer for the valuable suggestion. We agree that other signalling pathways may be also involved in the process of vascular restenosis. In the revised manuscript, we have discussed this important issue (please see the paragraph marked in red on page 18 in the Discussion session, line 561-567).
5. In the Discussion, it would be interesting to provide some more translational perspective. Besides its role in body weight control leptin also acts as a vasoactive hormone (ref Trovati M, Doronzo G, Barale C, Vaccheris C, Russo I, Cavalot F. Leptin and vascular smooth muscle cells. Curr Pharm Des. 2014;20(4):625-34). The expression of functional leptin receptors has been detected in VSMCs. Moreover, leptin modifies angiotensin II-induced vascular responses (ref Fortuño A, Rodríguez A, Gómez-Ambrosi J, Muñiz P, Salvador J, Díez J, Frühbeck G. Leptin inhibits angiotensin II-induced intracellular calcium increase and vasoconstriction in the rat aorta. Endocrinology. 2002 Sep;143(9):3555-60). These additional intracellular mechanisms can be mentioned in the Discussion to broaden the implications of the findings.
Response: Thank you for the constructive suggestion and meaningful comment. We agree with the reviewer that leptin is an important adipokine which may have a significant impact on the function of vascular system. However, the present study mainly focused on uncovering the effect and underlying mechanism of EP4 in the pathogenesis of restenosis. We would request to explore the crosstalk between EP4 and leptin/leptin receptor in vascular remodeling in a separate study in the future.
6. At the end of the Discussion, indicate limitations of the study; e.g. the work was performed only in males, the signalling may be altered in pathophysiological circumstances like obesity.
Response: Thank you for your suggestion. We have stated the limitations of our study. Please see the last paragraph of the Discussion section (marked in red, line 586-590).
Reviewer 2 Report
The authors nicely demonstrated on different levels (immunohistochemistry, well as on mRNA level and protein level) that EP4 promotes the proliferation and migration of VSMCs. The experiments and results support the hypothesis.
In the methods section 2.10. and 2.11.the dosage used of the agonist and antagonist should be mentioned . The statistical section 2.14. needs a major editing. If the authors compare more than two groups like in Figure 3 a 2-tailed students t test is not correct and which correction was done with the ANOVA. The First paragraph in the Discussion should be deleted.
Points for improvement:
- A number of three for some of the experiments is quite less.
- The authors should as well perform the experiments with different dosages of the EP4 agonist and antagonist to see if there is a dose dependant effect
- Are there any correlations or observations in humans? Are there any association with NSAID used in patients after eg undergoing a coronary angiography?
Author Response
- In the methods section 2.10. and 2.11.the dosage used of the agonist and antagonist should be mentioned.
Response: Thanks for your suggestion. The dosages used in our study were illustrated in the Figure legends.
- The statistical section 2.14. needs a major editing. If the authors compare more than two groups like in Figure 3 a 2-tailed students t test is not correct and which correction was done with the ANOVA.
Response: Thank you for your valuable comments. Indeed, as stated in this section, comparisons between 2 groups were tested by 2-tailed Student’s t test, while comparisons among multiple groups were made by ANOVA. In Figure 3A and other figures in which more than two groups were involved, the comparisons were made by ANOVA. We clarified this issue in the session of Statistics in the revised manuscript.
- The First paragraph in the Discussion should be deleted.
Response: Thank you for your kind suggestion. We have deleted the paragraph.
Points for improvement:
- A number of three for some of the experiments is quite less.
Response: We appreciate the reviewer’s valuable comment. In some western blot assays, the sample number is small. However, since we repeated each experiment for three times (in three separate experiments), we believe that the results are correct and reliable. The reviewer’s point is well taken, we will increase the number of the experiments in all future studies.
- The authors should as well perform the experiments with different dosages of the EP4 agonist and antagonist to see if there is a dose dependant effect
Response: We highly appreciate the reviewer’s meaningful comments. In fact, in our preliminary experiments, we had performed dose-dependent and time-dependent studies to select an appropriate dose and time period for the EP4 agonist and antagonists, which were utilized in the formal experiments shown in the manuscript.
- Are there any correlations or observations in humans? Are there any association with NSAID used in patients after eg undergoing a coronary angiography?
Response: Thanks for your constructive comments. We agree with the reviewer that it would be better to conduct a research or an observation in humans. However, so far we have not obtained sufficient numbers of vascular samples from patients to perform such correlative study. However, in the present study, we created a mouse overexpressing human EP4 gene (VSMC-hEP4 Tg) for our research and found overexpression of human EP4 accelerated restenosis in the VSMC-hEP4 Tg mice, suggesting that EP4 might enhance vascular restenosis in humans as well.
Regarding the association of the NSAID use with angiography, in an open-label randomized controlled study Koo B, et al. reported that the selective cyclooxygenase-2 (COX-2) inhibitor celecoxib can markedly reduce in-stent late luminal loss in patients receiving the implantation of paclitaxel-eluting stents (Lancet. 2007 Aug 18. PMID: 17707751). However, the underlying mechanism of this potential clinical benefit is unclear. Our present study suggests that celecoxib may attenuate in-stent neointima formation by blocking COX-2-mediated PGE2 biosynthesis, leading to reduced EP4-elicited TN-C expression.